# Association between Exposure to Particulate Matter during Pregnancy and Multidimensional Development in School-Age Children: A Cross-Sectional Study in Italy

**DOI:** 10.3390/ijerph182111648

**Published:** 2021-11-05

**Authors:** Paolo Girardi, Silvia Lanfranchi, Libera Ylenia Mastromatteo, Massimo Stafoggia, Sara Scrimin

**Affiliations:** 1Department of Developmental and Social Psychology, University of Padova, Via Venezia 8, 35131 Padova, Italy; silvia.lanfranchi@unipd.it (S.L.); liberaylenia.mastromatteo@phd.unipd.it (L.Y.M.); sara.scrimin@unipd.it (S.S.); 2Department of Statistical Sciences, University of Padova, Via Cesare Battisti 241, 35121 Padova, Italy; 3Department of Epidemiology, Lazio Regional Health Service, ASL Roma 1, Via Cristoforo Colombo, 112, 00147 Rome, Italy; m.stafoggia@deplazio.it

**Keywords:** air pollution, development, cognitive, communicative, robust model, multidomain assessment

## Abstract

Air pollutants can potentially affect the development of children. However, data on the effect of exposure to air pollution during pregnancy and developmental outcomes in school children are rare. We investigated the link between prenatal exposure to particulate matters smaller than 10 microns (PM_10_) and the development of school-age children in multiple domains. Cross-sectional data were collected in Italy between 2013 and 2014. Children aged between 5 and 8 years (n = 1187) were assessed on cognitive, communication, socio-emotional, adaptive, and motor developmental domains using the Developmental Profile 3 questionnaire. The monthly average concentration of PM_10_ during the entire fetal period was linked to the municipality of residence of the children. The increase in the prenatal PM_10_ was associated with a decrease in the cognitive score during the second (+13.2 µg/m^3^ PM_10_ increase: −0.30 points; 95%CI: −0.12–−0.48) and third trimesters of pregnancy (−0.31 points; 95%CI: −0.11–−0.50). The communicative domain was also negatively influenced by PM_10_ increases in the second trimester. The development of cognitive and communicative abilities of children was negatively associated with the exposure to PM_10_ during the period of fetal development, confirming that exposure to air pollution during pregnancy can potentially hinder the development of the brain.

## 1. Introduction

The combustion of fossil fuels in the last century has resulted in a progressive air pollution increase, which affects human health [1]. Epidemiological studies provide a clear link between air pollutant exposure and health status [2]. Exposure to air pollution during the early stages of life, from conception to the first two years, can have significant effects on children’s health. The growth of children’s bodies and organs occurs through complex developmental processes that can be easily disrupted, or even altered, when exposed to pollutants. Studies linking exposure to air pollutants to neurobehavioral, neuropsychiatric, and cognitive functioning in children reported mixed results [3], mainly due to the limited number of studies, their small sample size, and methodological constraints [4,5,6]. Nevertheless, the biological link between air pollution and development is supported by the fact that the central nervous system is extremely vulnerable due to the long time needed for its complete development [7]; the child is more susceptible to pollutants and has a reduced clearance of toxicants compared to adults [8].

Exposure to air pollutants is particularly detrimental during pregnancy for brain development [9,10,11,12,13], however it is also associated with immediate severe adverse health outcomes, such as fetal and infant mortality, low birth weight, preterm birth, intrauterine growth retardation, and birth defects [14]. Studies assessing the influence of prenatal exposure to air pollution on child development are limited and focused only on a specific domain. In a sample of children aged 6–11 years (*n* = 1105; the United States), increases in nitrogen dioxide (NO_2_) and fine particulate (PM_2.5_) exposure during the third trimester were associated with a decrease in nonverbal IQ and visual-motor abilities [15]. A large study among six European countries (9482 children aged 1–6 years) found an association between NO_2_ exposures during pregnancy and reduced psychomotor development [3]. Studies involving homogeneous populations [16,17,18,19] or longitudinal evaluations [20,21,22] confirmed the significant impact of prenatal exposure to air pollution on behavioral and cognitive functioning in children. In particular, studies comparing results between prenatal and postnatal exposure confirmed that pregnancy is a target temporal window [22,23]. In addition, direct childhood exposure to air pollution (PM_2.5_) can be associated with a reduction in cognitive abilities [24] and psychomotor development [3,21], as well as behavioral and mental health problems [5,25,26]. In general, studies on exposure to particulate matter, in particular those considering particles smaller than 10 microns (PM_10_), can be of crucial importance; in fact, PM_10_ originated from multiple sources, such as vehicle traffic, industrial settings, and natural sources (i.e., dust, forest, and plants) [27], and it represents a shared indicator of environmental pressure [28].

A further and major lack in the literature is related to the fact that most studies focused only on some specific developmental domain; this does not allow to assess the effect of pollution on the whole child. We adopted a multidimensional approach assessing the development across five key areas: motor, adaptive behavior, social-emotional, cognitive, and communication. These domains are of paramount importance to picture a child’s general functioning and generate a developmental profile. The present exploratory study aims to assess the effects of exposure to PM_10_ during pregnancy on each developmental domain among a representative sample of school-age children living in a wide geographical area of Italy with different levels of air pollution. Importantly, with the aim of creating a picture as much complete as possible, a series of elements that play an important role in determining development (e.g., family socioeconomic status, composition, and parental characteristics) were also considered.

## 2. Materials and Methods

### 2.1. Study Population and Assessment of Development

In total, 1519 school-age children (5 to 8 years) residing in a large set of municipalities in Italy between 2013 and 2014 voluntarily took part in the study. Data were collected through an ad hoc questionnaire administered to the children’s mothers by trained interviewers. Selection criteria were applied which excluded from the study mothers whose children had genetic disorders, clinically diagnosed neurodevelopmental disorders, or severe health problems, and who were born very premature. The first part of the questionnaire was designed to collect socio-demographic information about the child (such as birth date, gender, nationality, and municipality of residence), family (such as the number and the age of siblings), and parents (type of job, grade of education, and nationality). The education level of the parents was coded in low (middle school or less), medium (high school), and high (degree or higher). To complete this first part of the questionnaire participants took approximately 5 min. To assess the developmental domains, the Developmental Profile 3 (DP-3) [29] was used. DP-3 assesses child development exploring cognitive, communication, socio-emotional, adaptive, and motor domains through parent reports of everyday behaviors. It consists of 180 items, divided in five scales, that describe a child’s normal behavior that is considered a developmental milestone in a particular developmental domain. The parents must state whether their child shows that particular behavior at a certain frequency. DP-3 can be administered as a semi-structured interview or as a questionnaire that is to be completed by parents with an active support of the interviewer. The type of compilation was chosen by the interviewer to increase the data quality based on the mother’s availability and the understanding of the Italian language. Participants took about 20–30 min to complete the DP-3, both when administered as an interview or as a questionnaire. Out of 1519 children, 1157 children were successfully linked with the temporal window of the available air pollution data (for further details, see the next section). The geographical distribution of the 1157 children at their residential municipality has been reported in the Supplementary Material (Appendix A). In our sample, 54.3% of parents were interviewed by a trained interviewer, and 45.7% self-filled the questionnaire. In the latter case, an interviewer was available in case clarifications on the questions were required. During the years 2013 and 2014, 33 interviewers performed a face-to-face interview or administered the questionnaire with a mean of 44.7 questionnaires per interviewer. DP-3 has shown good psychometric properties, both in its original version and in the Italian version [30] that has been used in this study. The test’s manual reports retain good internal consistency (with most of the values higher than 0.90), test–retest reliability (with values ranging from 0.87 and 0.99), and good inter-rater reliability (with values higher than 0.95).

### 2.2. Exposure to Air Pollution

The data on air pollution were collected using a spatiotemporal model, which combined data of air pollution monitoring stations with satellite retrievals and GIS-based spatial predictors [31,32]. Briefly, we adopted a three-stage approach where we calibrated the monitor-based data to satellite aerosol optical depth retrievals and other spatial parameters in the first stage. We filled gaps in the satellite parameters using atmospheric dispersion models in the second stage and predicted the daily mean PM_10_ concentrations for each km^2^ of Italy (2006–2015) in the third stage. The monthly average of PM_10_ between 2006 and 2009 was linked to each participant’s residential municipality during the month of birth and the 9 months prior to birth. For each child, we calculated the average PM_10_ concentrations during each trimester of pregnancy and overall.

### 2.3. Statistical Methods

Descriptive statistics of the characteristics of the children, parents, extent of exposure, and developmental scores have been summarized in tables employing position and variability statistics. Robust multiple linear Mixed-Effects (ME) regression models were used to study the relationship between the levels of PM_10_ concentration and the developmental level in the domains previously described (motor, adaptive behavior, socio-emotional, cognitive, and communication). Analyses were performed separately for each domain. The models were adjusted for a list of variables, gender, child’s age in months at the DP-3 test date, educational level of the mother/father, type of job of mother/father, number of siblings, and presence of older siblings, that were potentially associated with development. Through a record linkage on the municipality of residence, we considered in the analysis the deprivation index (year of reference: 2001) [33], which combines information at the residential level about low level of education, unemployment rate, lack of home ownership, single-parent family, and high population density. The deprivation index was categorized according to the quintiles of the Italian municipalities. Other indices derived from the 2011 Italian census, specifically the aging index (population >65 years/<14 years) and the residential mobility index (% population changed habitual residence in the last year/resident population), were also considered. The aging index was categorized in tertiles, while the residential mobility index was binary-transformed based on the median value. We employed cubic b-splines to model the age effect to consider the nonlinear contribution of age on the considered developmental domains. A number of basis equal to 7 (six knots equally spaced according to the age distribution at 60, 68.7, 75.0, 78.9, 83.8, and 93.1 months) was chosen as optimal as it minimized the BIC (Bayesian Informative Criterion) index in the General Developmental score. Finally, we evaluated the effect of PM_10_ concentration (the average recorded during the 9 months before birth and each trimester of pregnancy) on the development of children. The effects of the continuous PM_10_ changes on the outcomes were estimated for every 13.2 µg/m^3^ increase in the average of the PM_10_ concentration, which corresponds to the IQR calculated for the overall PM_10_ during the pregnancy period. We adopted a robust regression model to account for the presence of heavy tails, outliers, and/or influential observations and the non-bell-shaped marginal distribution of some outcome of interest (Figure 1). The robustness of the estimates was guaranteed by means of the Huber smooth M-estimator (default parameter k = 1.345, s = 10) that yields 95% efficiency for the mean. The model incorporated the presence of two random effects as random intercepts to consider the hierarchical structure of the data collection flow: the type of data collection (questionnaire or interview) and the interviewer’s name.

The influence of the population mobility on the results was tested by a sensitivity analysis, estimating a model that included an interaction term between the residential mobility index and the average prenatal PM_10_ concentration. The regression models were estimated by means of the R program (version 4.0) and robustlmm packages [34].

## 3. Results

Table 1 presented a description of the characteristics of the children participating in the study. We included a total of 1157 participants, predominantly male (52.8%) (average age: 76.4 months; standard deviation (SD) = 7.9 months). Almost half of them had an older sibling (43.1%). Only 37.8% of mothers were full-time employees, while 91.2% of the fathers were full-time employees. The educational level of the mothers was slightly higher than that of the fathers (level: 27.2% vs. 18.8%). The children and their parents were mostly Italian (96.7% children; 94.2% and 95.9% mother and father, respectively). Children were predominantly residents of northern Italy (71.5%). Data collection was performed mainly during the year 2014 (62.2%).

Analysis of the Italian deprivation index revealed that children living in areas governed by municipalities were served by good access to services and overall quality of life (deprivation index low or very low: 75.9%). The aging index was high with a median of 145.2% (IQR: 118.7–195.8), and a general low residential mobility was reported (median: 6.26%; IQR: 5.32–6.30).

Among the children under study, the median PM_10_ concentration was 20.2 µg/m^3^ (range min-max, 8.5–53.6 µg/m^3^, Inter Quartile Range (IQR): 13.2 µg/m^3^) during the 9 months of pregnancy. The PM_10_ concentration was higher in the Northern Italian municipalities in all the mentioned sub-periods (at birth, during the pregnancy, and in all the trimesters) when compared to the other areas of Italy (*p* < 0.001, Table 2).

The pairwise correlation between prenatal PM_10_ concentration and motor and adaptive behavior score was weak (Spearman’s rho: 0.138 and 0.150, respectively), but positive and statistically significant. Prenatal air pollution exposure and other developmental domains were marginally uncorrelated (Figure 2). Among domains, the Cognitive and Communication Scores reported the highest correlation (Spearman’s rho: 0.791).

In the ME regression models, the General Developmental Score reported a decrease of 1.03 points for each IQR increase in PM_10_ (13.2 µg/m^3^) during the period of pregnancy, considering several confounding variables (Table 3). The adjusted effect was mainly concentrated during the second trimester (β = −0.72 (95% CI: −1.38, −0.06). The observed decrease in the General Developmental Score was explained by the negative effects of PM_10_ in the Cognitive (β = −0.73 (95% CI: −1.06, −0.41) and the Communication Score (β = −0.37 (95% CI: −0.66, −0.07).

As for the General Developmental Score, the effect related to the Cognitive and Communication Score was mainly attributed to the increase in PM_10_ in the second trimester (*p* = 0.001 and *p* = 0.018 for the cognitive and communicative domains, respectively). The Cognitive Score was also affected by a strong negative association with the PM_10_ concentration during the third trimester (β = −0.31 points per 1-IQR PM_10_ increase (95% CI: −0.50, −0.11)). Scores concerning the Motor, Adaptive Behavior, and Socio-emotional domains did not appear to be influenced by PM_10_ concentrations during the period of pregnancy. The model related to the Cognitive and Communication Score reported a satisfactory amount of variance explained (~50–60%) as well as that reported for the General Developmental Score.

The effect of PM_10_ concentrations on the considered developmental domains was adjusted for several covariates (Appendix A). In addition to the age, other important predictive characteristics considered to determine the overall development were the employment status of the father, the educational level of the mother, and the aging index at the municipality level. Foreign children reported a lower Motor Score with respect to Italian children. A decrease in the Adaptive Score was reported for the children living in low-deprived municipalities compared to those living in a medium deprived municipality.

Results from the sensitivity analyses validated the previously reported relationships between the General Development, Cognitive, and Communication Score. The results highlighted the presence of an inverse relationship between the Motor and Adaptive Scores and the municipalities at different residential mobility (Figure 3). The interaction between Residential Mobility (RM) and prenatal PM_10_ concentration included in a regression model was significant only for the Motor Score (*p* = 0.018), indicating that prenatal exposure to PM_10_ concentration may also influence the motor domain.

## 4. Discussion

The present study reports a significant negative association between PM_10_ exposure during pregnancy and overall development in school-age children living in different regions of Italy. PM_10_ exposure negatively correlated with cognitive and communication developmental domains. Interestingly, significant correlations were observed in children who were exposed to PM_10_ during the second trimester of gestation. No significant associations were found between the increase in the prenatal air pollution levels and variation in the Motor, Adaptive Behavior, and Socio-Emotional Scores in school-age children, indicating that the effects of PM_10_ may be limited only to certain developmental domains. However, results from sensitivity analyses revealed a potentially negative effect of exposure on motor development when residential mobility was considered.

The present results are in line with those of previous studies showing a negative correlation between the exposure to air pollutants during the third trimester of pregnancy (particularly PM_10_, PM_2.5_, and NO_2_), and cognitive development during infancy [35,36] and childhood [15,37,38].

The present study, which provided a global approach that considers all the most important developmental domains, extends the knowledge on the effects of prenatal exposure to pollution on the development of children in later stages, indicating a correlation between cognitive and communication development.

The results did not confirm previously reported results, which find a developmental delay in fine motor abilities [18] and behavioral functioning [19]. Hence, further studies are needed to assess the multiple developmental domains to better understand the relationship between exposure to air pollution and different developmental domains. However, results from sensitivity analysis suggest that further links may be discovered by setting up studies that consider other crucial variables such as residential history and daily mobility.

Researchers have also considered the time of the exposure (prenatal, postnatal, or childhood) reporting a growing body of evidence of adverse neurodevelopmental effects of combustion-related pollution during the prenatal window [39]. Our findings confirmed how the second trimester of pregnancy is particularly critical for future cognitive, communicative, and general development. The results reveal by what means exposure to air pollution during this time window can hinder development. In utero, the brain conformation and maturation begin as early as 6 weeks of pregnancy. In the second and third trimesters, the brain undergoes maximum development as a rapid increase in functional connectivity is observed [40] and the adult neuronal cell number is reached [41]. Different regions of the brain form at different times. Thus, the timing of gestational exposure to pollution and their severity may have a different long term impact on brain functioning after birth and the expression of particular developmental delays. The effect of PM_10_ levels on the cognitive domain was observed in both the second and third trimester of pregnancy. Thus, indicating that this period of rapid maturation the brain might particularly suffer from exposure to PM_10_. Few researchers examined the effect of trimester-specific windows of susceptibility. Recent literature supports the plausibility of an association between air pollution exposure (during the second and the third trimesters) on the cognitive and attentional domains [42]. Other studies suggested causal models consistent with a third-trimester association with the behavioral domain, in particular, the autism spectrum disorder [43,44]. These associations were explained by the toxicological effects of air pollution on brain development [45].

However, it is important to note that child development is determined by the continuous transactions between both individual and environmental factors. The effects of the combination of these multitude of factors is unique for each child. For example, brain development is a spatial and temporal continuous process that can be affected by specific vulnerability factors such as maternal nutrition, infection, and stress during pregnancy [46]. However, the presence of a family-supportive environment, breastfeeding duration, and the number of older siblings, can work as significant protective factors that help to mitigate the reported detrimental effects on children’s development [47].

In the present study, several possible confounding factors that might have an impact on child development were considered. Our results revealed a significant role of socioeconomic status on child development. Child development was specifically influenced by the maternal level of education, the job status of the father, and the aging index at the municipality level. There is a shared agreement that all these aspects play a significant role in determining children’s performance in all developmental domains [48]. However, even when these effects were controlled, exposure to PM_10_ significantly impacted the children’s cognitive and communicative functioning. This is of great importance as people living in heavily polluted areas often report greater social and economic deprivation than people living in less polluted areas. For these reasons, the air quality in poor neighborhoods should be controlled to protect children from a potential cumulative effect [49,50].

Our results support previous data that link exposure to air pollutants with low developmental levels in children under conditions of controlled variables such as socioeconomic status. We also report that exposure to PM_10_ during pregnancy does not affect all developmental domains equally. We conducted a standardized test that assessed child-functioning in a multitude of different domains by means of the same measure to present a complete developmental profile of the affected areas. This is particularly important when considering preventive and supportive measures directed to foster the developmental profile of children exposed to pollution during pregnancy.

Herein, we show that other developmental domains, for example, adaptive behavior, motor development, and socio-emotional functioning, do not seem to be equally affected. It is possible to hypothesize that although exposure to pollution might influence brain development, the development in these areas is significantly influenced by the experiences that the child gains during his early life. Of course, further studies need to be conducted to explore this hypothesis.

There are certain limitations of this study. Air pollution exposure was based on averaged PM_10_ concentrations at the municipality level. The lack of assessments that consider other pollutants as the NO_2_ and PM_2.5_ did not permit the attribution of the estimated effects to a specific source such as vehicle combustion, home heating, and industrial settings [27]. However, PM_2.5_ represents a significant proportion of PM_10_, and these two indicators are commonly correlated [27,28]. The attribution of the pollution at municipal level did not permit a consideration of the low-scale variability as well as the presence of important roads, highways, or industrial settings near the house or children’s school. As we considered in the analysis the long-term effects of air pollution, the use of monthly averaged PM_10_ concentration over a territory appears appropriate and justifiable. As the extent of air pollution was, on average, high in low-income areas and racial/ethnic minority areas, confounding socioeconomic status would tend to produce a bias away from the null. However, we adjusted all the parameters of analyses (parental education, employment status, race/ethnicity of the child, and deprivation index) as a proxy for wealth. The deviations in the results’ PM_10_ can be attributed to the categories not fully capturing the variation in the socioeconomic status or other related factors such as nutrition or quality of parent-child interactions. One of the limitations of the study includes the limited geographic coverage of the population that may affect the generalizability of results. In addition, we did not collect information about a series of individual characteristics such as the maternal and/or household smoking habit, stress events during pregnancy and at the birth, time spent outside by the mother, and the mother’s job with a risk for occupational exposure during pregnancy. In particular, the lack of information on the residential history may produce a bias in relation to the attributed air pollution. However, the potential bias might have been non-differential with respect to the outcomes, leading to the underestimation of the strength of observed associations [51]. As a support of that, this effect was confirmed by the results obtained from the proposed sensitivity analysis aimed to study the effect of the residential mobility index. A further indication of the goodness of our results was corroborated by the presence of developmental domains that were not associated with air pollution (i.e., adaptive and socio-emotional domains) as reported in the literature [52], constituting a negative control [53]. In the end, the plausibility of having omitted one reported confounder strong enough to drive the estimates related to air pollution exposure to zero was poor [54]. Furthermore, the analysis was enriched by a verification strategy that encompassed a sensitivity analysis and the use of robust regression models including the specification of random effects that helps to keep account of the sampling strategy.

## 5. Conclusions

Herein, we report the detrimental effect of exposure to PM_10_ during pregnancy on the cognitive and communicative functioning of school-age children. We assessed child development (using a standardized test) in a number of different domains that, when interconnected, are independent of one another. This multidimensional assessment allows us to better understand which specific developmental domains are more affected by air pollution. It should be noted that the present data do not allow us to conclude that the entire observed effect on children’s cognitive and communicative functioning is solely attributable to PM_10_ exposure in the second and third trimesters. As a matter of fact, it is more plausible that a mix of pollutants, for which PM_10_ is a marker, is indeed truly responsible for inducing systemic effects which affect the developing brain. More data should be recorded to better understand the effect of specific pollutants [55]. The PM_10_ concentration varied from region to region. The effects of PM_10_ concentration and the dose of exposure on child development on multiple domains were studied. Research should be conducted to assess the grade and the type of impact of pollution on cognitive functioning to identify ways to reduce health and developmental risks that this vulnerable and susceptible fraction of the population is subjected to.

## Figures and Tables

**Figure 1 ijerph-18-11648-f001:**
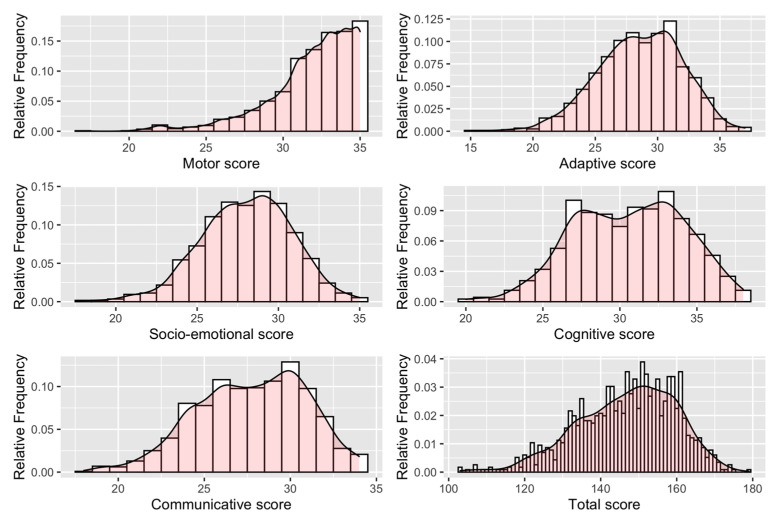
Distribution of the developmental score by domain of interest.

**Figure 2 ijerph-18-11648-f002:**
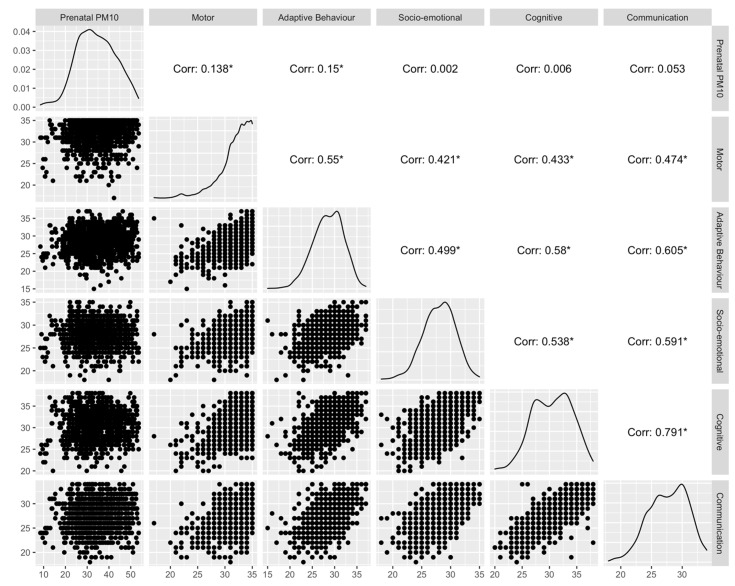
Pairwise marginal distribution and Spearman’s correlation between average prenatal PM_10_ concentration and developmental domains (non-parametric correlation test: * *p* < 0.05).

**Figure 3 ijerph-18-11648-f003:**
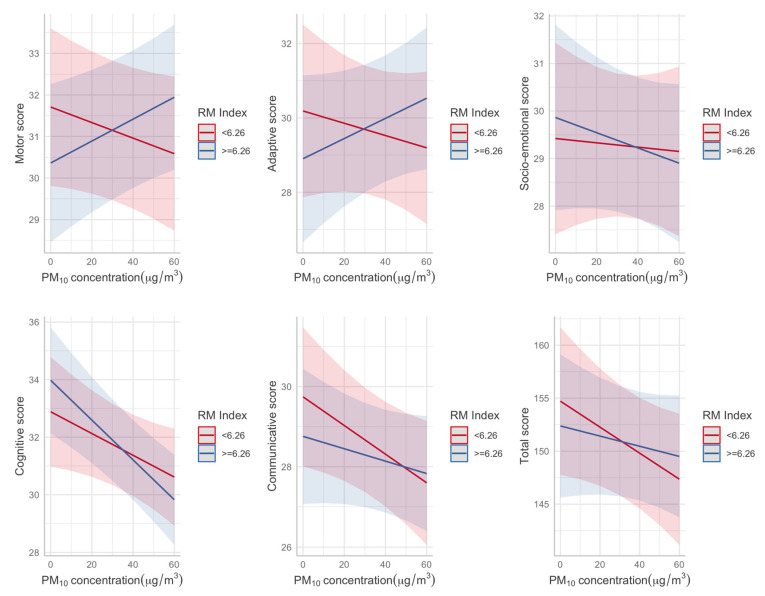
Predicted developmental scores by average prenatal PM_10_ concentration for different categories of Residential Mobility (RM) index.

**Table 1 ijerph-18-11648-t001:** Main characteristics of the children included in the study.

Characteristics	N = 1157
Gender [Male], *n* (%)	611 (52.8)
Age (months), Mean (SD)	76.4 (7.9)
Number of Siblings, Median (IQR)	1 (0)
Presence of older siblings, *n* (%)	499 (43.1)
Mother employment, *n* (%)	
Full time	437 (37.8)
Part-time	370 (32.0)
Housewife	300 (25.9)
unemployed/student	50 (4.3)
Father employment, *n* (%)	
Full time	1055 (91.2)
Part-time	102 (8.8)
Mother educational level, *n* (%)	
Low	267 (23.1)
Medium	575 (49.7)
High	315 (27.2)
Father educational level, *n* (%)	
Low	374 (32.3)
Medium	566 (48.9)
High	217 (18.8)
Children nationality, *n* (%)	
Italian	1119 (96.7)
Non-Italian	38 (3.3)
Mother nationality, *n* (%)	
Non-Italian	67 (5.8)
Italian	1090 (94.2)
Father nationality, *n* (%)	
Non-Italian	48 (4.1)
Italian	1109 (95.9)
Zone of residence, *n* (%)	
North	827 (71.5)
Center	138 (11.9)
South and Islands	192 (16.6)
Year at the interview, *n* (%)	
2013	437 (37.8)
2014	720 (62.2)
Deprivation Score Index category, *n* (%)	
Very-low	466 (40.3)
Low	412 (35.6)
Medium	118 (10.2)
High	14 (1.2)
Very-high	147 (12.7)
Aging index category, *n* (%)	
Low [57.1–123]	388 (33.5)
Medium (123–188]	383 (33.1)
High (188–294]	386 (33.4)
Residential mobility index category, *n* (%)	
Low-Medium [3.27–6.26]	594 (51.3)
Medium-High (6.26–9.30]	563 (48.7)

**Table 2 ijerph-18-11648-t002:** PM_10_ concentration at birth and during different periods of pregnancy by zone of residence.

PM_10_ Concentration (μg/m^3^)	North (*n* = 827)	Center (*n* = 138)	South and Islands (*n* = 192)	*p*-Value *
At Birth, Median (SD)	36.1 (22.8)	27.1 (11.1)	23.9 (7.6)	<0.001
Prenatal average, Median (SD)	36.9 (11.8)	29.7 (9.8)	25.8 (5.0)	<0.001
1st trimester, Median (SD)	33.3 (19.6)	29.6 (9.1)	25.9 (8.4)	<0.001
2nd trimester, Median (SD)	31.6 (18.4)	28.0 (9.7)	26.7 (8.3)	<0.001
3rd trimester, Median (SD)	34.7 (21.2)	28.4 (10.3)	24.6 (7.7)	<0.001

* Kruskal-Wallis test.

**Table 3 ijerph-18-11648-t003:** Adjusted * estimates of the coefficients for PM_10_ concentration, overall and for each trimester of pregnancy, for each domain of interest by robust ME models. Bold values denote statistical significance at the *p* < 0.05 level.

	**Motor Score**			**Adaptive Score**			**Socio-Emotional Score**		
**PM_10_** **(+13.2 μg/m^3^)**	$\hat{\beta }$	**95% CI**	***p*-Value**	$\hat{\beta }$	**95% CI**	***p*-Value**	$\hat{\beta }$	**95% CI**	***p*-Value**
Prenatal average	−0.00	−0.28–0.27	0.984	0.14	−0.25–0.52	0.494	−0.16	−0.50–0.19	0.373
Marginal R^2^/ Conditional R^2^	0.202/0.293			0.255/0.305			0.169/0.227		
1st trimester	0.04	−0.14–0.22	0.655	0.19	−0.05–0.42	0.117	−0.04	−0.25–0.18	0.744
2nd trimester	−0.03	−0.20–0.14	0.728	−0.08	−0.30–0.14	0.465	−0.09	−0.28–0.10	0.371
3rd trimester	0.01	−0.16–0.19	0.902	0.16	−0.07–0.39	0.177	−0.00	−0.21–0.21	0.983
Marginal R^2^/ Conditional R^2^	0.203/0.292			0.261/0.301			0.168/0.225		
	**Cognitive Score**			**Communicative Score**			**General Score**		
**PM_10_** **(+13.2 μg/m^3^)**	** $\hat{\beta }$ **	**95% CI**	***p*-Value**	** $\hat{\beta }$ **	**95% CI**	***p*-Value**	** $\hat{\beta }$ **	**95% CI**	***p*-Value**
Prenatal average	−0.73	−1.06–−0.41	**<0.001**	−0.37	−0.66–−0.07	**0.015**	−1.03	−2.19–0.12	0.080
Marginal R^2^/ Conditional R^2^	0.562/0.597			0.474/0.508			0.485/0.514		
1st trimester	−0.02	−0.22–0.17	0.820	0.02	−0.16–0.20	0.836	0.27	−0.45–0.99	0.468
2nd trimester	−0.30	−0.48–−0.12	**0.001**	−0.20	−0.36–−0.03	**0.018**	−0.72	−1.38–−0.06	**0.032**
3rd trimester	−0.31	−0.50–−0.11	**0.002**	−0.11	−0.29–0.07	0.228	−0.27	−0.98–0.45	0.464
Marginal R^2^/ Conditional R^2^	0.567/0.600			0.475/0.509			0.486/0.516		

* Adjusted by age (7 basis of splines), gender, employment status of the mother, employment status of the father, educational level of the mother, educational level of the father, nationality of the children, number of siblings, presence of older siblings, quintile of deprivation index score, tertiles of old-age index.

## Data Availability

The dataset generated during and/or analyzed during the current study is available from the corresponding author upon reasonable request.

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
