# Peer review of "Association between Exposure to Particulate Matter during Pregnancy and Multidimensional Development in School-Age Children: A Cross-Sectional Study in Italy"

_ijerph, 2021, doi:10.3390/ijerph182111648_

Round 1
Reviewer 1 Report
I thank the authors for conducting an important study on prenatal exposure to air pollution and its long-term effects on children's development. The authors have meticulously selected variables for the study and the hypothesis is strong. However, the study can be strengthened by adding or clearing few points.
Major comments:
- The major issue of the current study is it is not clear why the authors have chosen to study only PM10 exposure. Exposure to the majority of air pollutants, i.e. PM10, PM2.5, Ozone, Sulfur dioxide and Nitrogen oxides, occur concomitantly. Without studying other air pollutants, particularly PM2.5, I am not sure the authors can attribute childhood developmental delays to only PM10 exposure. This is the major limitation of the study. I would encourage the authors to at least study PM2.5. If it is not possible, at least, give a strong rationale in the introduction to study only PM10 exposure. Also, add that as a limitation of the study. I agree with the conclusion of the study. Please add this information to the introduction and also in the limitation of the study.
- Similar to the above comment, the authors did not study the postnatal PM10 exposure into childhood. The cumulative, as well as acute high dose of PM10 exposures, has been shown to have negative effects on various acute and chronic diseases. I would encourage the authors to also study the cumulative prenatal PM10 exposure and its effect on the outcome. Also, please comment on childhood exposure.
- Another important variable is exposure to maternal and/or household smoking. Please comment on that.
- The authors did not comment on the gestational age and birthweight of the children. Prematurity and low birth weight are associated with developmental delays. This reviewer suggests including only term babies with normal birth weight for the study to circumvent this problem. This is an extremely important variable to control and the authors may get surprising results after including these variables. On the other hand, prenatal exposure to air pollutants is associated with premature delivery, growth-restricted neonates. Therefore, it is better to exclude those children.
- In the method section, please define variables. What do low, medium, and high education for mothers and fathers mean?
Minor comments:
- Page 1, line 43 does not read well. The introduction is very long and written in one paragraph. I would break down the entire introduction into 3 paragraphs. Maternal exposure to metal is irrelevant and should be removed. Lines 55 to 63 can be moved to the discussion. The introduction should only have problems that the authors are studying, objectives and what is know, and what are they doing differently than previously published.
- Method - "In our sample, 54.3% of parents were interviewed by a trained interviewer, and 45.7% self-filled the questionnaire. In the latter case, an interviewer was available in case clarifications on the questions were required." It is not clear why only half of the parents were interviewed. Was it due to the language barrier? How did the authors determine who is going to get interviewed and who is allowed to have a self-filled questionnaire?
- Multiple punctuation errors. Please correct them.
Author Response
Reviewer 1
I thank the authors for conducting an important study on prenatal exposure to air pollution and its long-term effects on children's development. The authors have meticulously selected variables for the study and the hypothesis is strong. However, the study can be strengthened by adding or clearing few points.
Authors’ response: We thank the reviewer for appreciating our work and offering us the opportunity to revise our manuscript, following their insightful and constructive comments.
Major comments:
- The major issue of the current study is it is not clear why the authors have chosen to study only PM10 exposure. Exposure to the majority of air pollutants, i.e. PM10, PM2.5, Ozone, Sulfur dioxide and Nitrogen oxides, occur concomitantly. Without studying other air pollutants, particularly PM2.5, I am not sure the authors can attribute childhood developmental delays to only PM10 exposure. This is the major limitation of the study. I would encourage the authors to at least study PM2.5. If it is not possible, at least, give a strong rationale in the introduction to study only PM10 exposure. Also, add that as a limitation of the study. I agree with the conclusion of the study. Please add this information to the introduction and also in the limitation of the study.
Authors’ response: We agree with the suggestion, however the PM2.5 concentration is not available for the considered period of study. In addition in Italy PM10 and PM2.5 reported a good correlation due to common sources. We have added a sentence for better explain the choice and a sentence in the limitation part of the conclusions section.
- Similar to the above comment, the authors did not study the postnatal PM10 exposure into childhood. The cumulative, as well as acute high dose of PM10 exposures, has been shown to have negative effects on various acute and chronic diseases. I would encourage the authors to also study the cumulative prenatal PM10 exposure and its effect on the outcome. Also, please comment on childhood exposure.
Authors’ response: We considered the cumulative PM10 exposures as the overall average PM10 exposures during the pregnancy (the average exp. = cumulative exp./9 months).
- Another important variable is exposure to maternal and/or household smoking. Please comment on that.
Authors’ response: We agree with the suggestion. Smoking habit of the mother may affect in several ways the development of the children. Unfortunately this information is not available. We have added a sentence as a further limitation.
- The authors did not comment on the gestational age and birthweight of the children. Prematurity and low birth weight are associated with developmental delays. This reviewer suggests including only term babies with normal birth weight for the study to circumvent this problem. This is an extremely important variable to control and the authors may get surprising results after including these variables. On the other hand, prenatal exposure to air pollutants is associated with premature
delivery, growth-restricted neonates. Therefore, it is better to exclude those children.
Authors’ response: Rigid selection criteria were adopted before conducting the interviews or administering the questionnaires. Specifically, children with genetic or diagnosed disabling diseases as well as very preterm children have been excluded. Accordingly we have added a sentence in the methods section to better explain this selection and a sentence of the introduction to report as prenatal exposure to air pollutants is also associated to adverse effects during pregnancy.
- In the method section, please define variables. What do low, medium, and high education for mothers and fathers mean?
Authors’ response: We have added a sentence to define the education level of the parents.
Minor comments:
- Page 1, line 43 does not read well. The introduction is very long and written in one paragraph. I would break down the entire introduction into 3 paragraphs. Maternal exposure to metal is irrelevant and should be removed. Lines 55 to 63 can be moved to the discussion. The introduction should only have problems that the authors are studying, objectives and what is know, and what are they doing differently than previously published.
Authors’ response: We have broken down the introduction into 3 paragraphs, removing the sentence about maternal exposure to heavy metals and the lines between 55 and 59 that are now considered for the discussion. The introduction was completely revised accordingly with your comments.
- Method - "In our sample, 54.3% of parents were interviewed by a trained interviewer, and 45.7% self-filled the questionnaire. In the latter case, an interviewer was available in case clarifications on the questions were required." It is not clear why only half of the parents were interviewed. Was it due to the language barrier? How did the authors determine who is going to get interviewed and who is allowed to have a self-filled questionnaire?
Authors’ response: Parents could choose to participate in the study by replying to a structured interview or filling a questionnaire, but under the supervision of a trained interviewer. Parents with language difficulties were addressed towards an interview. We have added a sentence in the methods section to better explain it.
- Multiple punctuation errors. Please correct them.
Authors’ response: We have read the paper correcting the punctuation errors.
Reviewer 2 Report
Reviewer’s report
Title: Association between exposure to fine particulate matter during pregnancy and multidimensional development in school-aged children: a cross-sectional study in Italy
Date: 26 Sept 2021
Manuscript ID: 1399617
Reviewer’s report: In the manuscript titled “Association between exposure to fine particulate matter during pregnancy and multidimensional development in school-aged children: a cross-sectional study in Italy” the authors evaluated the link between exposure to fine particulate matter in during fetal development and development of school-aged children. The authors found a link between increased matter and a hindering effect on brain development.
Comments:
The main concerns about this study are: 1- data backs to 7-15 years ago and it’s not up to date AND 2- a lot of interfering factors could affect the results of the study which have been missed.
Introduction:
- The five domains that were used to assess development (physical, adaptive behavior, social-emotional, cognitive, and communication) were outlined. It would be beneficial to, briefly, discuss why these five were selected. What resources did you use to select these as your domains?
- Page 2, lines 48-49, please address the delays in which aspect of development do the authors mean? Also, it should be specified in the rest of the manuscript.
Materials and methods:
- Why were only mothers selected as interviewees? Please discuss the rationale behind selected one parent for interviewing.
- Please clarify why some participants were interviewed by a trained interviewer and others filled out the questionaries independently. It could affect the results of the questionnaire.
- There are some other factors that can interfere with the children's development like, parents' age, nutrient status of the mother during pregnancy, pregnancy complications such as preeclampsia, placenta previa and etc, which have been missed in this manuscript.
- How long took to answer the questionnaire?
- Were there any benefits for the participants in the study?
- The authors should provide more details about the DP-3 questionnaire.
- How did the authors determine the exposure time of each pregnant mother to air pollution? All mothers did not spend the same time outside.
- Some of the pregnant mothers might receive more pollutant from their workplace, that makes bias in the interpreting of the results.
- Did the authors consider any birth defects and APGAR scores of the children who participated in the study?
Statistical methods:
- it could be valuable to your readers to include the factors in the “deprivation index”, as this is valuable information that you considered during your data analysis.
Figures:
- Figure 1. This figure is clear and communicates important information.
- Figure 2: This figure is clear and well-made but does not convey information that is vital to the readers. It is more interesting to explore the link between PM10 and development, and less vital to understand how PM10 changes with location.
Discussion:
- one limitation that should be considered is that these children may be continually exposed to fine particulate matter outside of the gestational period that may have lasting impacts on their development. For example, if a child has high levels of PM10 as a fetus, and then is continually exposed to the PM10 throughout his life, how can you show that differences in development are more likely liked to in utero exposure? Please consider adding this to your discussion.
- Conclusion: It is mentioned that it is most likely many pollutants (represented by PM10) that impact child development. Please consider adding this to the discussion as what these specific pollutants may be.
- A list of key abbreviation definitions should be provided.
- Most of the references are old and need to be updated.
Author Response
Title: Association between exposure to fine particulate matter during pregnancy and multidimensional development in school-aged children: a cross-sectional study in Italy
Date: 26 Sept 2021
Manuscript ID: 1399617
Reviewer’s report: In the manuscript titled “Association between exposure to fine particulate matter during pregnancy and multidimensional development in school-aged children: a cross-sectional study in Italy” the authors evaluated the link between exposure to fine particulate matter in during fetal development and development of school-aged children. The authors found a link between increased matter and a hindering effect on brain development.
Comments:
The main concerns about this study are: 1- data backs to 7-15 years ago and it’s not up to date AND 2- a lot of interfering factors could affect the results of the study which have been missed.
Authors’ response: data were collected between 2013 and 2014, 7 years ago, but long time has been required to collect all the data and to clean them before to perform the analysis. We agree that a certain amount of confounders were not collected and we have revised the limitation paragraph of the discussion.
Introduction:
- The five domains that were used to assess development (physical, adaptive behavior, social-emotional, cognitive, and communication) were outlined. It would be beneficial to, briefly, discuss why these five were selected. What resources did you use to select these as your domains?
Authors’ response: The five considered domains represent the main domains that define child development. As a consequence, with slightly different names, these are the domains that are typically assessed in comprehensive tests to assess child development.
- Page 2, lines 48-49, please address the delays in which aspect of development do the authors mean? Also, it should be specified in the rest of the manuscript.
Authors’ response: In the new version of the introduction the mentioned sentence has been removed.
Materials and methods:
- Why were only mothers selected as interviewees? Please discuss the rationale behind selected one parent for interviewing.
Authors’ response: Mothers were selected as they usually represent in the Italian culture the primary caregivers and hence the ones knowing child development in a more precise and accurate way. Moreover, given that some questions were related to pregnancy and the first years of life of the child we thought that mothers would be better informed. In addition, in order we chose to have just one informant to have a more homogeneous sample.
- Please clarify why some participants were interviewed by a trained interviewer and others filled out the questionaries independently. It could affect the results of the questionnaire.
Authors’ response: Parents could choose to participate in the study replying to a structured interview or filling a questionnaire but even under the supervision of a trained interviewer. The variable type of data collection was kept into account in the regression model as a random intercept.
- There are some other factors that can interfere with the children's development like, parents' age, nutrient status of the mother during pregnancy, pregnancy complications such as preeclampsia, placenta previa and etc, which have been missed in this manuscript.
Authors’ response: We agree with the reviewer's comments. Genetic disorders, diagnosed disabling diseases as well as very preterm children represent an exclusion criterion for the study. However, the presence of pregnancy complications was not considered in the questionnaire. This has now been added to the limitations section of the manuscript.
- How long took to answer the questionnaire?
Authors’ response: Participants took about 5 minutes to complete the socio-demographic part of the questionnaire and about 20-30 minutes to answer the questions of the Developmental Profile, both administered as an interview and as a questionnaire. This information has been added to the method section.
- Were there any benefits for the participants in the study?
Authors’ response: The participation is completely voluntary and no benefit will be expected.
- The authors should provide more details about the DP-3 questionnaire.
Authors’ response: In the paper DP-3 main characteristics are now specified as follow: DP-3 assesses child development exploring cognitive, communication, socio-emotional, adaptive, and motor domains through parent reports of everyday behaviors. It consists of 180 items, divided in 5 scales, that describe a child’s normal behavior that is considered a developmental milestone in a particular developmental domain. The parents have to state whether their child shows that particular behavior at a certain frequency. DP-3 can be administered as a semi-structured interview or as a questionnaire that is to be filled by parents with an active support of the interviewer. DP-3 has shown good psychometric properties both in its original version and in the Italian version [3128] that has been used in this study. The test’s manual reports retain good internal consistency (with most of the values higher than 0.90), test-retest reliability (with values ranging from 0.87 and 0.99), and good inter-rater reliability (with values higher than 0.95).
- How did the authors determine the exposure time of each pregnant mother to air pollution? All mothers did not spend the same time outside.
Authors’ response: We agree with the comment. An important indicator is the balance of time spent outside and on the other side, the presence of other potential pollutants (formaldehyde, heating by wood combustion, etc…) inside home may be harmful. However this information is not available. We have revised the limitation paragraph of the discussion to highlight this limitations.
- Some of the pregnant mothers might receive more pollutant from their workplace, that makes bias in the interpreting of the results.
Authors’ response: In the analysis we have considered the job of the mothers as a confounders in order to keep into account of that. We agree that a good solution can be to combine the exposure at home with that at job, but it is not possible. We have added that as a limitation in the discussion.
- Did the authors consider any birth defects and APGAR scores of the children who participated in the study?
Authors’ response: This is a very relevant point. Children with impairing birth defects or with diagnosed neurodevelopmental disorders were not included in the study. We agree that the APGAR score would have been a good variable to take into account, however this index is not something that mothers commonly known or remember. For this reason it was not possible to collect this data.
Statistical methods:
- it could be valuable to your readers to include the factors in the “deprivation index”, as this is valuable information that you considered during your data analysis.
Authors’ response: we have included information about the factors at the basis of the deprivation index in the method section.
Figures:
- Figure 1. This figure is clear and communicates important information.
Authors’ response: Thank you for the appreciation.
- Figure 2: This figure is clear and well-made but does not convey information that is vital to the readers. It is more interesting to explore the link between PM10 and development, and less vital to understand how PM10 changes with location.
Authors’ response: We have reported a new figure between the marginal correlation between average prenatal PM10 and developmental domain scores.
Discussion:
- one limitation that should be considered is that these children may be continually exposed to fine particulate matter outside of the gestational period that may have lasting impacts on their development. For example, if a child has high levels of PM10 as a fetus, and then is continually exposed to the PM10 throughout his life, how can you show that differences in development are more likely liked to in utero exposure? Please consider adding this to your discussion.
Authors’ response: We agree with the comment. We have added a sentence in the introduction about the importance of the prenatal exposure. We also added a new sentence in the discussion to better highlight this important point.
- Conclusion: It is mentioned that it is most likely many pollutants (represented by PM10) that impact child development. Please consider adding this to the discussion as what these specific pollutants may be.
Authors’ response: We have added a new sentence in the discussion.
- A list of key abbreviation definitions should be provided.
Authors’ response: According to the journal guidelines, we have defined each abbreviation the first time that it appears in the text.
- Most of the references are old and need to be updated.
Authors’ response: the introduction and conclusions section now are supported by new updated series of references.
Reviewer 3 Report
Thank you for the opportunity to read your paper titled “Association between exposure to fine particulate matter during pregnancy and multidimensional development in school-age 3 children: a cross-sectional study in Italy.” This paper presented finding on the relationship between exposure to fine PM during pregnancy and multiple domains of childhood development. The study found that increased exposure to prenatal PM during pregnancy was associated with a decrease in cognitive score and communicative ability in children. I believe these findings are impactful. Here are my recommendations for the author(s)-
- Line 35-36: Is this statement referring to data examining the relationship between functioning in children and exposure to pollutants after birth or in utero?
- Line 37: What were the “mixed and inconsistent” findings reported?
- Line 39: The focus of this paper is exposure to pollutants in utero. Given that these are two distinct time periods in childhood development with distinct levels of vulnerability, why is the discussion of exposure to pollutants after birth relevant?
- Introduction: Some epidemiological data would provide more context to this discussion on the significance of the problem being explored.
- Line 42&43: This sentence needs to be revised to enhance clarity.
- The introduction should be broken down into different paragraph to enhance readability. In its current form, the presentation of extant literature is not well-organized and is hard to follow. The author(s) should work on better synthesizing extant literature so the reading can clearly follow the narrative and identify where the gap exists.
- Line 291: The author states “..it is important to note that child development is determined by individual and environmental factors.” However, this is not discussed in the limitations paragraph as a limitation of the study. The fact that several individual and environmental confounders are not measured in this study is a notable limitation.
- Line 320-321: What evidence exists to support this hypothesis?
Author Response
Thank you for the opportunity to read your paper titled “Association between exposure to fine particulate matter during pregnancy and multidimensional development in school-age 3 children: a cross-sectional study in Italy.” This paper presented finding on the relationship between exposure to fine PM during pregnancy and multiple domains of childhood development. The study found that increased exposure to prenatal PM during pregnancy was associated with a decrease in cognitive score and communicative ability in children. I believe these findings are impactful. Here are my recommendations for the author(s)-
Authors’ response: We thank the Reviewer for appreciating our work and offering us the opportunity to revise our manuscript, following their insightful and constructive comments.
- Line 35-36: Is this statement referring to data examining the relationship between functioning in children and exposure to pollutants after birth or in utero?
Authors’ response: In the new version of the introduction this sentence has been deleted.
- Line 37: What were the “mixed and inconsistent” findings reported?
- Authors’ response: The introduction section has been revised and this sentence has been fragmented to better explain the mixed findings and inconsistent results reported by the cited manuscripts.
- Line 39: The focus of this paper is exposure to pollutants in utero. Given that these are two distinct time periods in childhood development with distinct levels of vulnerability, why is the discussion of exposure to pollutants after birth relevant?
- Authors’ response: We agree with the comment. The introduction now presents mainly studies about prenatal exposure. The exposure after birth was also considered, but just a term of comparison only in studies considering both exposures.
- Introduction: Some epidemiological data would provide more context to this discussion on the significance of the problem being explored.
- Authors’ response: We agree with the suggestion. We hope that the new version of the introduction meets the required improvement.
- Line 42&43: This sentence needs to be revised to enhance clarity.
Authors’ response: The sentence has been modified to enhance clarity.
- The introduction should be broken down into different paragraph to enhance readability. In its current form, the presentation of extant literature is not well-organized and is hard to follow. The author(s) should work on better synthesizing extant literature so the reading can clearly follow the narrative and identify where the gap exists.
Authors’ response: Following the suggestion we have broken down the introduction into 3 paragraphs organizing the contents in a more suitable way.
Line 291: The author states “..it is important to note that child development is determined by individual and environmental factors.” However, this is not discussed in the limitations paragraph as a limitation of the study. The fact that several individual and environmental confounders are not measured in this study is a notable limitation.
Authors’ response: We agree with the comment. We have enriched the limitation section to cover all the limitations, in particular those at individual level.
- Line 320-321: What evidence exists to support this hypothesis?
Authors’ response: Authors’ response: Most of the studies conducted in developmental psychology show the importance of the environment as well as of previous experiences for the development of the child (Bronfembremner, 1979; Sameroff, 2009). Hence, while exposure to air pollutants can affect several domains of child development, it is also true that other aspects intervene in the process as either protective (Bornstein MH, & Bradley RH, 2014) or additional risk factors (Blair, & Raver, 2012; Gottfried et al., 2003). For example, we can expect that a child growing up in a caring and supportive family, in a high socio-economic environment that offers repeated occasions for learning (e.g., parents expose the child to books, museums visits) to function fairly well in most domains despite being exposed to air pollution. On the contrary, children growing up in a more at-risk environment might display greater delays due to exposure pollution.
Round 2
Reviewer 1 Report
I thank the authors for responding to the majority of comments. However, I still think that introduction is awfully long. There are repetitions of ideas in the first paragraph itself. I would advise cutting the introduction in half. Readers will lose interest if it is this long. As I mentioned earlier, the introduction should be concise and point towards knowledge gaps in the research area, objectives of the study, and how the study is filling those knowledge gaps.
Author Response
Dear reviewer,
Accordingly with the suggestion, the introduction section was revised explaining the point in a concise way and deleting redundant information and/or sentences.
Kind regards